# Lightly Boron-Doped Nanodiamonds for Quantum Sensing Applications

**DOI:** 10.3390/nano12040601

**Published:** 2022-02-10

**Authors:** Masfer Alkahtani, Dmitrii K. Zharkov, Andrey V. Leontyev, Artemi G. Shmelev, Victor G. Nikiforov, Philip R. Hemmer

**Affiliations:** 1King Abdulaziz City for Science and Technology (KACST), Riyadh 11442, Saudi Arabia; 2Institute for Quantum Science and Engineering, Texas A&M University, College Station, TX 77843, USA; prhemmer@exchange.tamu.edu; 3FRC Kazan Scientific Center of RAS, Zavoisky Physical-Technical Institute, 420029 Kazan, Russia; dzharkov@list.ru (D.K.Z.); mailscrew@gmail.com (A.V.L.); sgartjom@gmail.com (A.G.S.); vgnik@mail.ru (V.G.N.); 4Department of Electrical and Computer Engineering, Texas A&M University, College Station, TX 77843, USA

**Keywords:** boron-doped nanodiamonds, diamonds, nitrogen-vacancy, quantum sensing

## Abstract

Unlike standard nanodiamonds (NDs), boron-doped nanodiamonds (BNDs) have shown great potential in heating a local environment, such as tumor cells, when excited with NIR lasers (808 nm). This advantage makes BNDs of special interest for hyperthermia and thermoablation therapy. In this study, we demonstrate that the negatively charged color center (NV) in lightly boron-doped nanodiamonds (BNDs) can optically sense small temperature changes when heated with an 800 nm laser even though the correct charge state of the NV is not expected to be as stable in a boron-doped diamond. The reported BNDs can sense temperature changes over the biological temperature range with a sensitivity reaching 250 mK/√Hz. These results suggest that BNDs are promising dual-function bio-probes in hyperthermia or thermoablation therapy as well as other quantum sensing applications, including magnetic sensing.

## 1. Introduction

Fluorescent bulk/nanodiamond crystals have attracted special attention in several important areas of application due to their optical properties, surface chemistry, and biocompatibility [1]. These applications include quantum information [2], advanced bio-sensing and drug delivery [1,3], optical temperature sensing [4,5,6], hyper-polarized magnetic resonance imaging (MRI) [7], nanoscale imaging down to the single protein level [8,9,10,11], and advanced materials diagnostics, especially for magnetic materials and superconductors [12]. The most interesting and useful color center in fluorescent nanodiamonds (FNDs) is the negatively charged color center (NV), which combines exceptional optical and spin properties at both low and room temperatures, making it a leading candidate for the above-mentioned applications [1].

Exploring quantum sensing using the NV color center in BNDs is of special interest due to the recent successful implementation of BNDs as an anticancer agent via hyperthermia/thermoablation therapy [13]. This study reports that BNDs can selectively heat the local environment, for example, in tumor cells, without heating the surrounding tissue by exciting it with NIR lasers (808 nm) in the tissue transparency window. This makes BNDs preferable to other nanodiamonds, such as crushed high-pressure and high-temperature (HPHT) and detonation nanodiamonds.

In order to make full use of BNDs in such an important application, it is essential to have an internal temperature sensor. The NV color center has demonstrated this ability, but boron doping in diamonds is known to quench the optical properties of the NV [14]. In this study, we report optical temperature and magnetic field sensing using the NV center in lightly boron-doped nanodiamonds. This study will open the door for the successful dual-functions of the BND applications as anticancer agents in hyperthermia/thermoablation therapy applications.

## 2. Materials and Methods

### 2.1. Sample Preperation

As received, lightly boron-doped nanodiamonds (~0.5 wt% of Boron) from Adamas Nanotechnologies were cleaned in a boiling mixture of nitric and sulfuric acids (HNO_3_:H_2_SO_4_ = 1:1, 120 °C), which is known to etch graphitic carbon from ND surfaces selectively. Afterward, the air was oxidized at 550 °C for 10 min to remove possible sp^2^ carbon on the surface, which can block ion implantations and quench the NV center luminescence in the BNDs. The cleaned BNDs were deposited on a quartz slide for ion implantation. To obtain a high density of NV centers in the BNDs, we implanted the samples with nitrogen at 50 keV energy with a dose of 2 × 10^12^ ion/cm^2^, followed by annealing in a vacuum at 750 °C for 30 min to mobilize vacancies towards nitrogen atoms in order to form the NV centers.

### 2.2. Flourecense and ODMR Spectra

To analyze the fluorescence and optically detected magnetic resonance (ODMR) spectra of the fluorescence nanodiamonds (FNDs), we designed and built a confocal laser scanning microscope. This confocal microscope was equipped with a high magnification microscope objective (100×, NA = 0.8, model number: LMPlanFL N, Olympus, Tokyo, Japan), multi-color lasers, and an integrated microwave system. The FND samples were attached to a microwave board and placed on the confocal setup. FND samples were then scanned in the x–y directions by a green (532 nm) laser (max power = 150 mW) using Thorlabs GVS 212 Galvano scanners (Newton, NJ, USA). The fluorescence spectra were collected through the same microscope objective and analyzed with a custom-made spectrometer equipped with a starlight camera (Trius camera model SX-674, Starlight Xpress Ltd., Bottle Lane, UK) and a photon counter (Hamamatsu photon counter model number H7155-21, Hamamatsu Photonics UK Limited, Welwyn Garden City, UK). For the ODMR, the microwave (MW) frequencies were swept over a specific range (for example, from 2700 MHz to 3000 MHz), and the fluorescent counts were plotted vs. MW frequency.

### 2.3. Rabi Oscillation Measurments

A 1 μs green laser pulse polarized the NV center and was followed by microwave pulses, with varying time duration t, at a fixed frequency (corresponding to the transition frequency between the m_s_ = 0 and one of the m_s_ = ±1 sublevels). Finally, a green laser pulse was applied to read out the NV center’s state and to record the Rabi oscillation spectrum [11].

### 2.4. Spin Lifetime Measurments (T_1_)

We used a 1 µs laser pulse to optically polarize the NV center into the m_s_ = 0 ground spin sublevel (^3^A_2_ state). Then, the NV defect was kept in the dark for a time, τ, causing the system to relax towards a mixture of states, m_s_ = 0, ±1. Finally, a second laser pulse was applied to read out the final electron spin population and to measure the NV center spin relaxation time (T_1_) [11].

### 2.5. Hahn-Echo Measurements

From the Rabi oscillations spectrum, we determined the pulse durations of π/2 and π pulses needed for the subsequent Hahn-echo measurements. Then, following a first green initialization laser pulse, three resonant microwave pulses π/2–π–π/2 were applied. The NV center electron spin accumulated a phase proportional to the amplitude of the oscillating magnetic field, acting along the NV center defect axis between these pulses. Finally, a second 532 nm laser pulse was applied to read out the final spin state of the NV center at the end of the measurement [11].

## 3. Results and Discussion

Lightly boron-doped nanodiamonds (~0.5 wt% of Boron), with an average size of 50–70 nm as a suspension in deionized water (DI), were purchased from Adamas Nanotechnologies. The BNDs were synthesized by a bottom-up HPHT approach. Prior to the experiments, the BNDs were cleaned in a boiling mixture of nitric and sulfuric acids (HNO_3_:H_2_SO_4_ = 1:1, 120 °C), which is known to etch the graphitic carbon from the ND surfaces selectively. Afterward, the acid-cleaned BNDs were air oxidized at 550 °C for 10 min to remove any remaining sp^2^ carbon on the surface, which can block ion implantation and quench the NV center luminescence.

The absorption spectrum of the cleaned BNDs (1 mg/mL) suspended in DI water was recorded, as illustrated in a. The absorption curve of the BNDs showed a peak of around 800 nm, as well as diamond and water absorption peaks in the ultraviolet (UV) and near-infrared (NIR) bands, respectively. The BNDs’ absorption around the wavelength of 800 nm was in good agreement with previously reported literature [15]. The heating ability of the BNDs under a continuous 808 nm laser irradiation was verified using two samples of undoped NDs and BNDs. For this, 2 mL each of undoped NDs and BNDs, at a concentration of (1 mg/mL) in DI water, were placed in two quartz vials (1 cm × 1 cm) for a laser heating experiment. Under continuous irradiation of 808 nm with a laser diode of an intensity of 50 W·cm^−2^ for 20 min, the time-dependent temperature rise was recorded, as shown in Figure 1b; a temperature rise of 15 °C was observed in the BNDs’ solution under an 808 nm laser illumination for 20 min, in contrast to 2 °C observed in the undoped NDs’ solution under the same laser irradiation for the same period of time. This is in agreement with prior work identifying BNDs as promising anticancer agents, as reported in [13].

Next, after verifying the heating ability of the lightly doped BNDs, it was important to investigate whether the BNDs could still function for quantum sensing applications, especially for temperature sensing [1]. For the optical characterization of NV color centers in BNDs, we designed a homebuilt multi-color confocal scanning microscope equipped with several continuous wave (CW) laser diodes (532 nm and 808 nm), as shown in Figure 2a. The excitation lasers were focused through a long working distance NIR microscope objective (100×) with NA = 0.8 (Olympus, model number: LMPlanFL N, Olympus, Tokyo, Japan). The fluorescence of the NV color center in the BNDs was collected and analyzed using a photon counter and a homebuilt spectrophotometer. We investigated the photoluminescence spectra of a thin layer of the BNDs’ crystals containing fluorescent NVs, which was prepared as follows: a droplet of 1 mg/mL of the BNDs mixed with polyvinyl alcohol (PVA) at a 1:1 *w*/*w* ratio, spin-coated on a piece of quartz to avoid undesired agglomerated emitters. The sample was then attached to a microwave board, which, in turn, was attached to a compact heater for optical temperature calibration. Figure 2b demonstrates a weak NV center fluorescence of the received BNDs (before ion implantation and annealing). It was clear that the boron acceptors in the BNDs transferred many of the negatively charged, NV^−^, centers into its neutral charge state, NV^0^, which degrades the spin-dependent optical properties [14].

To overcome this problem, we irradiated the BNDs via nitrogen implantation. Specifically, nitrogen at an energy of 50 keV was implanted with a dose of 2 × 10^12^ ion/cm^2^, followed by annealing in a vacuum at 750 °C for 30 min, as illustrated in Figure 2c. After irradiation and annealing, the BNDs were placed on the confocal microscope. Using a green laser (532 nm, 200 µW), we found a high density of bright fluorescent spots uniformly distributed on the BNDs’ sample. The optical fluorescence spectra collected from most of these spots show the signature of the NV center with NV^0^ and NV^−^ zero-phonon lines peaking at 575 nm and 637 nm, respectively and as illustrated in Figure 2d. The key result here is that the implanted nitrogen creates substitutional nitrogen (P1) centers (plus extra NVs), which act as donors, bringing up the Fermi level of the BNDs enough to make the NV^−^ charge state stable. However, the NIR absorption of the ionized boron acceptor is not degraded because a higher Fermi level actually increases the ionization fraction.

Proceeding, it was important to investigate the spin-sensing properties of the NV centers in the BNDs, since this is essential for quantum sensing applications, such as the nanoscale temperature and magnetic fields. For this, we first investigated the fluorescence contrast of the optically detected magnetic resonance (ODMR). Briefly, ODMR presents as a decrease in NV^−^ fluorescence when a microwave excitation is scanned over a ground- state spin transition involving the m_s_ = 0 and m_s_ = +/−1 levels in the triplet ground state, as illustrated in Figure 3a [11,16,17]. Typically, the fluorescence change is a maximum of about 30% for single NVs and 10% for ensembles, where this value is reduced to about half when there is a line splitting [18]. The left and right dotted circles in Figure 3a demonstrate how the zero-field splitting of the NV center varies depending on the temperature and magnetic field changes.

Furthermore, it was important to calibrate the temperature sensitivity of the NV in the BNDs. The NV senses temperature via temperature-dependent transition frequencies from the S = 0 to S = +/−1 spin levels. As shown in Figure 3a, the average splitting among S = 0, 1 sublevels is given by D(T), which is due to spin-orbit interactions and the diamond crystal field [17], where D(T) = 2.87 GHz at room temperature. This splitting depends on the diamond lattice expansion, which in turn depends on the local temperature, resulting in reduced spin-transition frequencies as the temperature increases. Temperature calibration was performed over a temperature range (298–330 K) using a precise heating stage. The observed linear dependence of the frequency shift of the ODMR vs. the temperature gave a slope of dM/dT = −72 kHz/K, which approximately agrees with the NV temperature sensitivity in non-boron-doped NDs reported in [6,19,20].

To demonstrate the multifunctional heating and temperature sensing with BNDs, we used an 808 nm laser as a heating source. Figure 3b shows ODMR spectral shifts for two representative temperatures of 298 K and 320 K, corresponding to the laser heating with 808 nm at intensities of 1 and 125 W/cm^2^, respectively. From these data, we calculated the thermal sensitivity of our BNDs using the thermal sensitivity equation derived in [19],
(1)$$\eta =\frac{4\sqrt{3}}{9}\frac{\Gamma }{C}{\left(\frac{\mathrm{dM}}{\mathrm{dT}}\right)}^{-1}\frac{1}{\sqrt{\mathrm{DI}}}$$
where η is the thermal sensitivity in K/√Hz, Γ is the ODMR linewidth, C is the contrast, dM/dT is the temperature slope (defined above), D is the room temperature value of the NV center zero-field splitting D(T), and I is the counting rate. By using the inhomogeneous ODMR width for Γ = 17 MHz, the ODMR contrast C = 4%, and the photon count rate of about 3.3 × 10^8^ photon/sec during ODMR measurements, we estimated the sensitivity of the NV center in BNDs to be 250 mK/√Hz.

In addition to optical temperature sensing, the NV center in BNDs can also sense small changes in local magnetic fields. Due to symmetry, the m_s_ = ±1 sublevels of the ground state are nearly degenerated at zero magnetic field (B = 0), resulting in a single resonance line (except for a small zero-field splitting) appearing in the ODMR spectrum, as illustrated in Figure 3c. An external magnetic field increase lifts the degeneracy of m_s_ = ±1, leading to the appearance of two separate lines, where the separation increases as the axial magnetic field increases, as shown in Figure 3c [21]. For more precise sensing applications, the NV spin longitudinal relaxation time, T_1,_ and spin coherence time, T_2,_ should be as large as possible [11,20,22]. In order to measure these, Rabi oscillations were first investigated to determine their ability to coherently manipulate the NV center’s electronic ground spin state, as described in detail in our previous work [11]. Figure 4a illustrates a multi-cycle Rabi oscillation between m_s_ = 0 and m_s_ = ±1 states of the NV center’s ground state. Pulsed microwave and optical excitations were then used to measure the NV spin longitudinal relaxation time, T_1_, and the spin coherence time, T_2,_ giving 370 µs and 5 µs, respectively, and as shown in Figure 4b,c.

## 4. Conclusions

In this work, we demonstrated multifunctional optical temperature control and sensing using the nitrogen-vacancy (NV) center in boron-doped nanodiamonds (BNDs) with an 808 nm heating laser in the tissue transparency window. In order to accomplish this, it was necessary to add nitrogen donors (via implantation) to the BNDs to stabilize the desired NV^−^ charge state and to create more NVs. Significantly, this co-doping did not reduce absorption of the ionized donors needed for the heating laser. Moreover, the sensing properties of these NVs were not degraded by the presence of the boron acceptors. This was verified by measurements of the spin longitudinal relaxation time, T_1,_ and the spin coherence time, T_2_. Additionally, local magnetic field sensing with the BNDs was briefly demonstrated. Finally, our past work on NVs in a nitrogen-doped diamond shows that it is also possible to excite the NV with a red laser ~637 nm in order that both lasers can be in the tissue transparency window [23]. These results open the door to the implementation of BNDs in hyperthermia and thermoablation therapy and enable future quantum sensing applications with this material.

## Figures and Tables

**Figure 1 nanomaterials-12-00601-f001:**
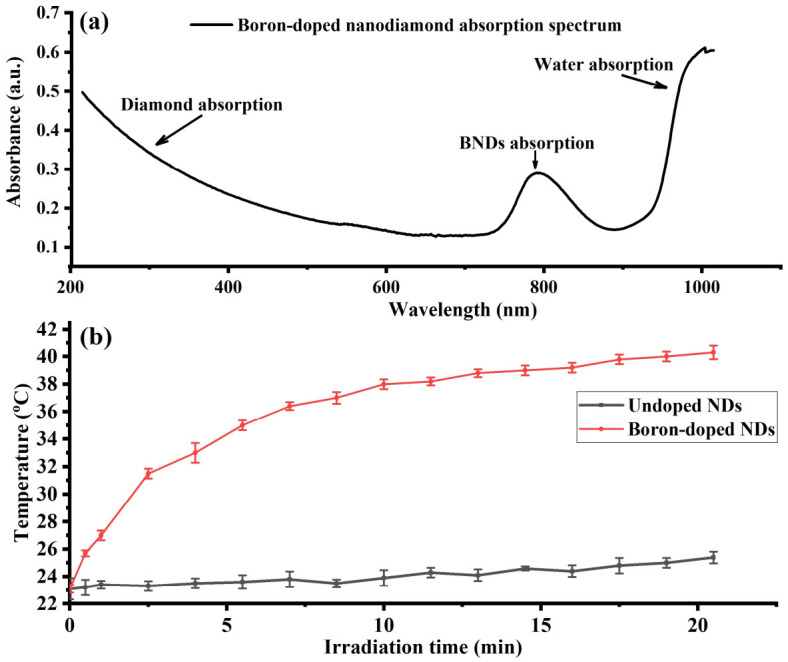
(**a**) Boron-doped nanodiamonds’ absorption spectrum recorded over the UV–VIS–NIR regions. BNDs show a relatively good absorption around the wavelength of 800 nm. (**b**) Comparison of temperature rise of 1 mg/mL of undoped NDs and BNDs dispersed in 2 mL of distilled water under 808 nm laser irradiation at 50 W·cm^−2^ for 20 min.

**Figure 2 nanomaterials-12-00601-f002:**
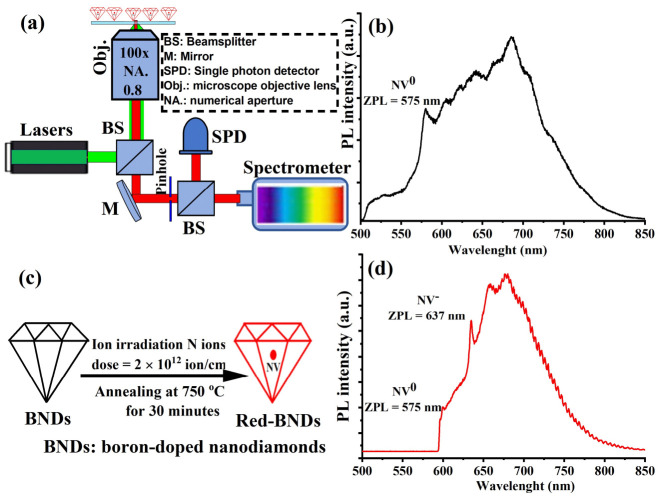
(**a**) An illustration of a custom-made confocal microscope designed and equipped with green and NIR lasers for optical quantum sensing measurements of boron-doped nanodiamonds (BNDs). (**b**) Photoluminescence optical spectrum of BNDs before irradiation. The PL spectrum shows a weak intrinsic NV center spectrum where the negatively charged (NV^−^) center is partially quenched by the presence of boron atoms. (**c**) An illustration of converting boron-doped nanodiamonds (BNDs) to more fluorescent BNDs after nitrogen implantation and annealing. (**d**) The NV spectrum in the irradiated BNDs shows much weaker neutrally charged (NV^0^) center compared to NV^−^ because the implanted nitrogen creates substitutional nitrogen (P1) centers, which act as donors, bringing up the Fermi level of the BNDs enough to make the NV^−^ charge state stable and dominant. The NV^−^ showed a clear PL spectrum with zero-phonon lines (ZPL) of NV0 and NV- peaked at 575 nm and 637 nm, respectively.

**Figure 3 nanomaterials-12-00601-f003:**
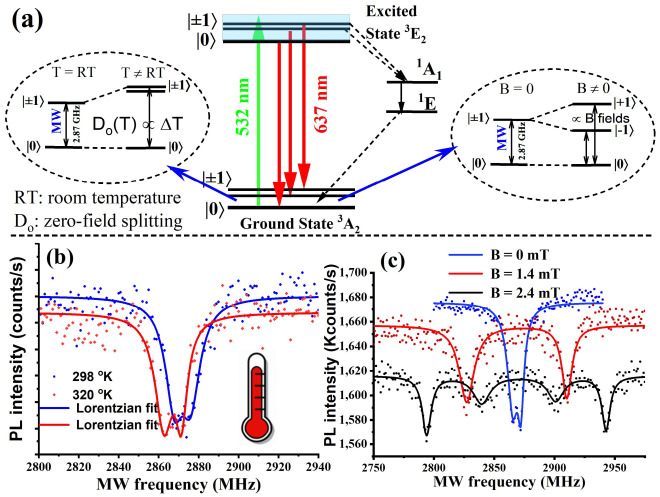
(**a**) An illustration of the energy diagram of the NV center’s ground spin states in the BNDs. The left and right dotted circles in Figure 3 (**a**) demonstrate how the zero-field splitting (D_0_)of the NV center varies depending on the temperature (T) and magnetic field (B) changes. RT symbol in (**a**) (left inset) stands for room temperature. (**b**) Optically detected magneto resonance (ODMR) spectrum of the NV center in BNDs at two different temperatures, 298 K and 320 K. (**c**) ODMR spectrum splitting due to different magnetic field values (0, 1.4, and 2.4 mT).

**Figure 4 nanomaterials-12-00601-f004:**
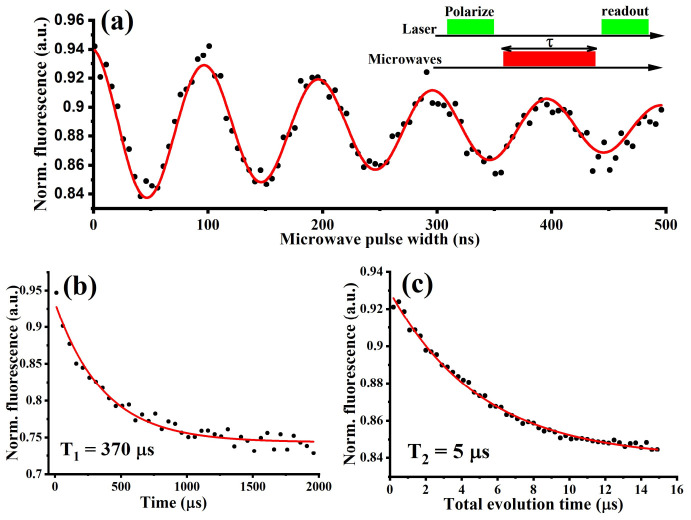
(**a**) The reported BNDs show clear Rabi oscillations between m_s_ = 0 and m_s_ = ±1 states. (**b**) longitudinal relaxation time T_1_ of the NV center. (**c**) The NV center spin coherence time (T_2_) measure using Hahn-echo pulse sequence.

## Data Availability

The data presented in this study are available from the corresponding author upon request.

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
