# Peer review of "Lightly Boron-Doped Nanodiamonds for Quantum Sensing Applications"

_nanomaterials, 2022, doi:10.3390/nano12040601_

Round 1
Reviewer 1 Report
The MS deals with an important and timely issue of the application of nanodiamonds (NDs) in medicine and diagnostics.
The authors exploit the possibility of local heating of Boron-doped ND by NIR light and employ the well-known thermometry based on NV color centers to assess its effect. To avoid adverse action of Boron admixture on NV signals caused by issues like fluorescence quenching, they use controlled implantation of Nitrogen ions followed by the sample annealing.
This clever strategy results in NDs with a multi-functionality: thermometry and magnetometry with NV, and local heating thanks to B absorption.
The MS is scientifically sound, well written, contains most of the relevant bibliography, and presents novel and conclusive results. I am therefore pleased to recommend its publication.
Before that happens, however, the authors should improve some Figures:
- In Fig.1b the error bars or a note in the figure caption on the size of the error bars should be added.
- 2b and 2c should be presented with the same scale and aligned vertically to enable the comparison of the relevant spectra. It looks as it the small amplitude of the NV0 component is caused mainly by the application of a cutoff filter. This point needs to be clarified.
- The meaning of the “RT” symbols in the left broken-line insert is unclear. It should be clearly defined in the figure caption.
Author Response
Referee 1’s comments
1) The authors exploit the possibility of local heating of Boron-doped ND by NIR light and employ the well-known thermometry based on NV color centers to assess its effect. To avoid adverse action of Boron admixture on NV signals caused by issues like fluorescence quenching, they use controlled implantation of Nitrogen ions followed by the sample annealing.
This clever strategy results in NDs with a multi-functionality: thermometry and magnetometry with NV, and local heating thanks to B absorption.
The MS is scientifically sound, well written, contains most of the relevant bibliography, and presents novel and conclusive results. I am therefore pleased to recommend its publication.
Response: We thank the Referee for his/her valuable time for reviewing, kindly giving us constructive suggestions to improve our manuscript, and positively recommending our manuscript for publication in Nanomaterials.
2) In Fig.1b the error bars or a note in the figure caption on the size of the error bars should be added.
Response:
We have added error bars to figure 1(b).
3) 2b and 2c should be presented with the same scale and aligned vertically to enable the comparison of the relevant spectra. It looks as it the small amplitude of the NV0 component is caused mainly by the application of a cutoff filter. This point needs to be clarified.
Response:
We have done an appropriate vertical alignment to Figure 2(b and c). Also, we have clarified the small amplitude of the NV0 component in figure 2 caption as highlighted in red in the revised manuscript.
4) The meaning of the “RT” symbols in the left broken-line insert is unclear. It should be clearly defined in the figure caption.
Response:
The RT symbol is clearly defined in the figure 3 caption as highlighted in red in the revised manuscript.

Reviewer 2 Report
The paper by Alkahtani et al. describes the development and evaluation of BNDs into which additional NVCs were implanted by irradiating nitrogen ion beam to commercially available BNDs. The evaluation was performed very briefly in terms of sensing the temperature and the magnetic field. The manuscript is written clearly, and the topic matches well to the current journal. Therefore, I recommend to publish the paper after minor queries as below mentioned will be cleared.
- The manuscript describes the optical setup as "confocal". However, pinholes were not shown in Fig. 2a. I suggest to illustrate the pinholes for readers better understanding.
- The objective used is described as "objective (100X) with NA = 0.8 (Olympus)". However, the photograph shown in Fig. 2a is of different one. First, add the exact model number of the objective in the main text; i.e., not only the magnifications and NA. Second, amend Fig. 2a.
- 2d legend. The zero-phonon line is better be described as "637 nm" instead of "638 nm" as in the corresponding figure.
- L186-189 "The observed linear dependence of the frequency shift..." Please show the data with which the slope dM/dT = -72 kHz/K is determined. Alternatively, if this value is referring to the value published elsewhere, state so by citing references.
- Eq (1) Should the parentheses at the end of the equation be removed?
Author Response
Referee 2’s comment
1) The paper by Alkahtani et al. describes the development and evaluation of BNDs into which additional NVCs were implanted by irradiating nitrogen ion beam to commercially available BNDs. The evaluation was performed very briefly in terms of sensing the temperature and the magnetic field. The manuscript is written clearly, and the topic matches well to the current journal. Therefore, I recommend to publish the paper after minor queries as below mentioned will be cleared.
Response:
We thank the Referee for his/her valuable time for reviewing, kindly giving us constructive suggestions to improve our manuscript, and positively recommending our manuscript for publication in Nanomaterials.
2) The manuscript describes the optical setup as "confocal". However, pinholes were not shown in Fig. 2a. I suggest to illustrate the pinholes for readers better understanding.
Response:
We have added a pinhole drawing into the optical setup as illustrated in figure 2(a).
3) The objective used is described as "objective (100X) with NA = 0.8 (Olympus)". However, the photograph shown in Fig. 2a is of different one. First, add the exact model number of the objective in the main text; i.e., not only the magnifications and NA. Second, amend Fig. 2a.
Response:
We have added the exact model of the used objective as highlighted in blue in the revised manuscript. Also, we have amended figure 2(a) accordingly.
4) 2d legend. The zero-phonon line is better be described as "637 nm" instead of "638 nm" as in the corresponding figure.
Response:
We have fixed this issue throughout the revised manuscript as highlighted in blue.
5) L186-189 "The observed linear dependence of the frequency shift..." Please show the data with which the slope dM/dT = -72 kHz/K is determined. Alternatively, if this value is referring to the value published elsewhere, state so by citing references.
Response:
The calibration data are not shown in this manuscript because the slope value matches the slope value of the NV center in diamond previously published in the literatures. So, we citied the appropriate references as highlighted in blue in the revised manuscript in line (193).
6) Eq (1) Should the parentheses at the end of the equation be removed?
Response:
Unnecessary parentheses in eq(1) have been removed.

Round 2
Reviewer 2 Report
The authors clarified all the concerns that I had in the previous version, hence recommend to publish in the present form.